# Prediction Model of Soleus Muscle Depth Based on Anthropometric Features: Potential Applications for Dry Needling

**DOI:** 10.3390/diagnostics10050284

**Published:** 2020-05-07

**Authors:** Juan Antonio Valera-Calero, Ladislao Laguna-Rastrojo, Fernando de-Jesús-Franco, Eduardo Cimadevilla-Fernández-Pola, Joshua A. Cleland, César Fernández-de-las-Peñas, José Luis Arias-Buría

**Affiliations:** 1Department of Physiotherapy, Faculty of Education and Health, Universidad Camilo José Cela, Villanueva de la Cañada, 28692 Madrid, Spain; ecimadevilla@ucjc.edu; 2Escuela Internacional de Doctorado, Universidad Rey Juan Carlos, 28933 Alcorcón, Spain; 3Private Professional Practice, Madrid, Spain; ladislao.laguna@alumno.ucjc.edu; 4Department of Pharmacy, Faculty of Health Sciences, Alfonso X el Sabio University, Villanueva de la Cañada, 28691 Madrid, Spain; fdejesus@uax.es; 5Doctor of Physical Therapy Program, Department of Public Health and Community Medicine, Tufts University School of Medicine, Boston, MA 02155, USA; clelandj@franklinpierce.edu; 6Department of Physical Therapy, Occupational Therapy, Rehabilitation and Physical Medicine, Universidad Rey Juan Carlos, 28933 Alcorcón, Spain; cesar.fernandez@urjc.es (C.F.-d.-l.-P.); joseluis.arias@urjc.es (J.L.A.-B.); 7Cátedra Institucional en Docencia, Clínica e Investigación en Fisioterapia: Terapia Manual, Punción Seca y Ejercicio Terapéutico, Universidad Rey Juan Carlos, 28933 Madrid, Spain

**Keywords:** soleus muscle, ultrasound, depth, anthropometric, gender

## Abstract

This study was conducted to investigate if anthropometric features can predict the depth of the soleus muscle, as assessed with ultrasound imaging, in a sample of healthy individuals to assist clinicians in the application of dry needling. A diagnostic study to calculate the accuracy of a prediction model for soleus muscle depth, as assessed with ultrasonography, in the middle-third and distal-third of the calf, based on anthropometric features such as age, height, weight, body mass index (BMI), calf length, mid-third and distal-third calf girth, was conducted on 48 asymptomatic healthy subjects (75% male) involving a total of 96 calves. Multiple linear regression analyses were used to determine which variables contributed significantly to the variance in the soleus muscle depth at middle-third and distal-third of the calf by gender. Women were found to have a deeper soleus muscle than men (*p* < 0.001). Weight, height, BMI, and mid-third calf perimeter explained 69.9% of variance in men, whereas mid-third calf perimeter, calf length, height, and distal-third calf girth explained 73% of the variance in women of the distal-third soleus depth (*p* < 0.001). Additionally, mid-third calf girth and calf length explained 28.8% of variance in men, whereas mid-third calf perimeter, calf length, and weight explained 67.8% of variance in women of the mid-third soleus depth (*p* < 0.001). This study identified anthropometric features that predict soleus muscle depth, as assessed with ultrasound, in asymptomatic individuals, but these features are different in men and women. Our findings could assist clinicians in choosing the proper length of the needle to avoiding passing through the soleus during dry needling.

## 1. Introduction

The soleus is a wide and flattened muscle located in the posterior compartment of the calf deep to the gastrocnemius muscles. The soleus muscle originates from the soleal line and middle third of the posterior aspect of the tibia, the proximal fourth third of the posterior surface of the fibula, and the posterior aspect of the fibular head and inserts, with the gastrocnemius, into the posterior part of the calcaneus forming the Achilles tendon [1]. Chow et al. reported a 2 cm mean thickness of the soleus muscle, with males exhibiting greater muscle thickness in general [2]. The soleus muscle contributes, with the gastrocnemius muscles, to ankle plantar-flexion, but also provides postural control of the ankle, particularly in weight bearing, since it controls ankle dorsiflexion when acting eccentrically.

Any musculoskeletal impairment in the soleus muscle could lead to biomechanical changes in the ankle. One common muscular impairment observed in the lower extremity are myofascial trigger points (TrPs). A TrP is defined as a sensitive spot in a skeletal muscle taut band that is painful on stimulation (e.g., manual palpation, muscle contraction, needling), elicits referred pain, and induces motor disturbances [3]. Trigger points can be considered active or latent depending on their ability to reproduce pain symptoms. Latent TrPs are asymptomatic from a sensory point of view but they promote motor disturbances, including restricted range of motion, muscle weakness, or accelerated fatigability [4]. Previous studies have reported a prevalence of latent TrPs in the soleus muscle ranging from 15% to 25% in asymptomatic people [5,6]. Similarly, Bajaj et al. [7] observed a prevalence of 15–20% of latent TrPs in the soleus muscle in a sample of patients with lower limb osteoarthritis. There is preliminary evidence suggesting a potential relationship between latent TrPs in the soleus muscle and restricted ankle range of motion; therefore, its management seems highly relevant for some sports [8]. In fact, an adequate ankle range of motion is necessary for functional activities such as running, gait, or ascending–descending stairs [9].

Previous studies have investigated the effects of TrP manual therapy of the triceps surae, which includes the soleus muscle, in individuals with plantar heel pain [10], calf pain [11] or recreational athletes [8]. All studies reported positive results in pain and mobility outcomes with the application of manual therapy; however, it should be considered that manual access to the soleus muscle is only possible on the lateral portion due to its anatomical location. This is the reason why some authors had advocated for the use of dry needling for proper targeting of the soleus [12]. The American Physical Therapy Association (APTA) defines dry needling as “an intervention using a thin filiform needle that stimulates the skin, TrPs, muscle or connective tissue for the treatment of musculoskeletal pain disorders” [13]. This intervention consists of the insertion of a solid filiform needle into a muscle with the aim to inactivate potential TrPs [12]. Since the soleus is located deep in the gastrocnemius muscles, the use of a 40 or 50 mm needle is usually recommended for penetrating this muscle. In fact, the length of the needle is commonly determined based on patient’s anthropometric features. It is important to be aware of the anatomical location of the posterior tibial nerve through the posterior compartment of the calf behind the soleus muscle [12], because crossing throughout the soleus muscle with the needle could lead to potential trauma to the tibial nerve. The tibial nerve is a main branch of the sciatic nerve which arises at the apex of the popliteal fossa. When the nerve travels through the popliteal fossa, it passes deep in the tendinous arch of the soleus muscle, usually at the mid-line, to enter the posterior compartment of the calf. In the calf, the nerve runs downwards and medially to the mid-line to finally reach the posteromedial side of the ankle [14].

Although most adverse events with dry needling are minor [15], potential damage of a nerve is possible in some cases [16]. In fact, different strategies have been proposed to optimize the patient’s safety and relative risk when performing dry needling. For instance, Ball et al. [17] have recently suggested that patient positioning can help to reduce the risk of puncturing the femoral nerve when dry needling the iliacus muscle. Others proposed the use of ultrasound imaging to reduce the number of adverse events during needling interventions [18]. Nevertheless, the use of ultrasound imaging during dry needling is not always possible due to the orientation of the needle in relation to the muscle and the probe. Additionally, ultrasound equipment is operator-dependent and involves economic costs [19].

Ferrer-Peña et al. [20] have recently developed a prediction model, based on forearm anthropometric measures, determining the necessary needle length to prevent median nerve injury during the application of dry needling on the pronator teres. A similar procedure could help clinicians to determine the soleus depth for necessary minimizing the risk of puncturing the posterior tibial nerve. Therefore, the aim of this study was to investigate if anthropometric features can predict the depth of the soleus muscle, as assessed with ultrasound imaging, in a sample of healthy individuals. We hypothesized that a prediction model based on anthropometric features of the calf would accurately predict the depth of the soleus muscle and could assist during the application of dry needling interventions in this muscle. Since gender differences are found in the soleus [2], we analyzed the prediction model in men and women separately.

## 2. Methods

### 2.1. Study Design

A diagnostic study to calculate the prediction model accuracy for soleus muscle depth, as assessed with ultrasonography, in the mid-third and distal-third of the calf based on anthropometric features such as age, height, weight, body mass index (BMI), gender, calf length, half-third and distal-third calf girth, was conducted. This study followed the Standards for the Reporting of Diagnostic Accuracy Studies (STARD) guidelines and checklist [21].

### 2.2. Participants

Consecutive healthy volunteers recruited from local flyer announcements located at a private clinic of Alfonso X El Sabio University between January–February 2020 were invited to participate. To be eligible to participate, volunteers had to be between 18 and 55 years old and have no report of pain in the lower extremity the previous year. Exclusion criteria included a prior history of recurrent pain in the lower extremity; any pharmacological treatment affecting muscle tone, e.g., muscle relaxants, analgesics; prior history of lower extremity surgery; lumbar radiculopathy or myelopathy; and any medical condition such as tumour or fracture. The study was approved by the Institutional Ethics Committee of Alfonso X El Sabio University (UAX 10-01-2020). Participants signed written informed consent prior to their inclusion in the study.

Anthropometric data included age, gender, height, weight, calf dominance and BMI [22]. Further, calf length (measured from the popliteal fossa to the calcaneal tuberosity) and calf girth (assessed in the mid-third and distal-third distance of the calf length) were also calculated (Figure 1).

### 2.3. Soleus Ultrasound Procedure

Subjects were placed in the prone position with their knees extended and 0 degrees of ankle dorsiflexion for both anthropometric and ultrasound measurements. A rigid table was placed on the foot surface of participants to assist in maintaining the ankle position. All ultrasound measurements were conducted by a clinician with more than 10 years of experience using ultrasound imaging. Soleus depth was measured in the middle-third and the distal-third of the calf by using an ultrasound equipment Alpinion Ecube i8 (Gyeonggi-do, Korea) with a linear transducer E8-PB-L3-12T 3–12 MHz. The thickness measurement between the skin and the lowest limit of the soleus muscle was performed (Figure 2). The mean of three repeated trials was calculated for the main analysis.

### 2.4. Sample size Calculation

Based on the only previous similar study [20], a sample size of at least 65 calves could be considered as appropriate. If we considered our design as a prognostic study, a range from 10 to 15 subjects per potential predictor, with no more than five predictor variables, is usually recommended to develop an adequate sample size for prediction models and for avoiding overestimation of the results [23]. Based on this alternative calculation a sample size of 50 calves would be required given the maximum cut-off of five predictors included in the final model. Since we originally include eight potential anthropometric features in the model, we calculated a sample size of at least 80 calves.

### 2.5. Statistical Analysis

Data analysis was conducted with the Statistical Package for the Social Science (SPSS) Version 21 (Armonk, NY, USA) for Mac OS. Normal distribution of the data was analyzed using the Kolmogorov–Smirnoff test (*p* > 0.05). After checking the sample homogeneity, Student *t*-tests for independent samples were used to determine gender and dominance differences. Multiple linear regression analyses were used to determine which variables contributed significantly to the variance in the soleus muscle depth at the middle-third and distal-third of the calf.

First, a correlation analysis between soleus depth and anthropometric features was performed using Pearson’s correlation coefficients (*r*) for normal distributed variables. Values of <0.3 were considered a poor correlation, from 0.3 to 0.5 a fair correlation, 0.6 to 0.8 a moderate correlation, and >0.8 a strong correlation [24]. Those variables that were statistically significant (*p* < 0.05) were included in a stepwise multiple linear regression model to estimate the proportion of variance explaining soleus muscle depth. The Pearson correlation coefficients were also applied to identify multicollinearity and shared variance between the variables (defined as *r* > 0.80). All correlation analyses were conducted in men and women, separately.

Hierarchical regression models were conducted to determine those variables that contributed significantly to soleus depth at each point, separately in men and women. The significance criterion of the critical *F* value for entry into the regression equation was set at *p* < 0.05. The adjusted changes in *R*^2^ were reported after each step of the regression model to determine the association of each additional variable in the regression model.

## 3. Results

From a total of 50 volunteers responding to the announcement, two were excluded due to previous calf trauma. Forty-eight asymptomatic subjects (75% men) were included (*n* = 96 calves). Table 1 summarizes the demographic and anthropometric data of the sample, by gender and calf dominance (right). Men were younger, taller and heavier, and have had longer calf length and more superficial soleus muscles than women (all, *p* < 0.001). No significant differences by calf dominance were observed.

Table 2 describes Pearson’s correlation coefficients between anthropometric features and soleus depth. Soleus depth, either at distal-third or mid-third calf point, was positively correlated with anthropometric features, except age, in both men and women. In addition, significant correlations also existed among the independent anthropometric variables with no multicollinearity; therefore, each variable was included in the regression analyses.

Table 3 shows the hierarchical regression analysis conducted in this study in men. For distal-third soleus depth, BMI contributed 27.7% of the variance (*p* < 0.001), weight contributed an additional 29.1% (*p* < 0.001), height an additional 10.2% (*p* < 0.001), and mid-third calf perimeter the last 2.9% of variance (*p* = 0.008). When combined, anthropometric features explained 69.9% of the variance of distal-third soleus depth in men (*p* < 0.001). For mid-third soleus depth, mid-third calf girth contributed 15.4% of variance (*p* < 0.001) and calf length contributed an additional 13.4% (*p* < 0.001). When combined, anthropometric features explained 28.8% of the variance of mid-third soleus depth in men (Table 3).

Table 4 shows the hierarchical regression analysis in women. For distal-third soleus depth, mid-third calf girth contributed 49.8% of the variance (*p* < 0.001), calf length contributed an additional 4% (*p* = 0.038), height an additional 9.4% (*p* = 0.002), and distal-third calf girth 9.8% of the variance (*p* = 0.008). When combined, anthropometric features explained 73% of the variance in the distal-third soleus depth in women. For mid-third soleus depth, mid-third calf girth contributed 56.1% of variance (*p* < 0.001), calf length contributed an additional 11% (*p* < 0.001), and weight the last 0.7% (*p* = 0.013). When combined, anthropometric features explained 67.8% of the variance of mid-third soleus depth in women (Table 4).

## 4. Discussion

This study found some anthropometric feature predicting soleus muscle depth in asymptomatic individuals, but these features are different in men and women and also depend on the level (mid-third or distal-third of the calf) where the soleus is assessed. These findings could guide clinicians in the use of dry needling interventions by assisting the selection of the proper length of the needle to avoid crossing the soleus muscle and potentially damaging the tibial nerve.

This is the first study to develop a prediction model for calculating the anatomical depth of a muscle based on anthropometric and ultrasound measurements. Ferrer-Peña et al. [20] created a decision tree depending on forearm girth to recommend the needle size with a 100% safety of median nerve accidental puncture. This recommendation is acceptable if clinicians cannot use ultrasound equipment, but this model cannot determine muscle depth. This could result in needling approaches not considering muscle depth, such as that of the soleus, which could lead to an accidental nerve puncture of deeper neurovascular structures. The results of the current study support that the first parameter to consider to determine needle length is gender. We observed between-gender differences (mean of 5 mm) in soleus depth, with women exhibiting a greater muscle depth than men. This was a surprising finding since men tend to have greater average muscle thickness and greater absolute strength than women; however, some gender differences in soleus architecture have previously been observed [2]. In fact, a deeper soleus in women could be related to the fact that women exhibit more adipose tissue than men. This hypothesis is supported by one study showing that, although men are heavier than women, they have a thinner fat layer in the calf region [25].

The variables with higher impact determining soleus depth were also slightly different between men and women. For instance, weight, height or BMI were more relevant in men than in women, whereas calf girth and calf length were more relevant in women. This maybe also be related to gender differences in fatty distribution and volume [26]. It seems that anthropometric features related to muscle physical demand, such as weight, height or BMI, are more relevant in men [27], which are usually more physically active than women, whereas anthropometric features more associated to volume, such as calf girth and length, are more relevant in women. It is also important to determine that the variables associated with soleus depth are also slightly different depending on the area of the calf (mid-third or distal-third of the calf) where the soleus muscle is assessed. Nevertheless, our data identified a robust model for distal-third soleus depth (explaining 70% of the variance) in both men and women.

### 4.1. Clinical Implications

Perhaps the most important findings are the results from the hierarchical regression analysis used to complete the prediction model to predict the distal-third soleus depth in men and women, which explained variances of 70% and 73%, respectively. Clinicians can use, in the absence of ultrasound equipment, the measurement of BMI, weight, height and mid-third calf girth to determine the depth of the soleus muscle in men and mid-third calf girth, calf length, height and distal-third calf girth in women. The predictive model was only able to explain 28.8% of the variance in the mid-third of the calf in men with the variables of calf girth and calf length. However, in women the mid-third of the calf girth, calf length, and height explained 68.8% of the variance. These identified models could be of considerable value to clinicians who target the soleus muscle with dry needling techniques as this could assist with selecting appropriate needle length and, therefore, preventing tibial posterior nerve injuries. Nevertheless, calculation of all anthropometric features could be slightly time-consuming.

The results of the current study suggest that clinicians should use needles ranging from 25 to 40 mm in length for needling the soleus muscle with a posterior needle approach while keeping in mind that the muscle is deeper in women than in men. This information is clinically relevant since the tibialis posterior neurovascular package runs under the soleus muscle; therefore, using needles longer than 40 mm could be risk puncturing this muscle with a posterior approach. The current prediction model could help clinicians to identify patients for using short needles (25–30 mm) based on anthropometric features. Future cadaveric studies could help to further elucidate the validity of this model in other needling approaches of the soleus muscle, e.g., the medial one [12]

### 4.2. Limitations

Although the current study has shown promising results, potential limitations should be recognized. First, this prediction model was based on a sample of healthy subjects, not equally distributed between men and women. Larger sample sizes are needed to determine normative values of soleus muscle depth. Second, we only assessed the depth of the soleus muscle without considering the thickness of the muscle or the adipose tissue. It is possible that the gender differences found in the current study are due to these outcomes and should be addressed in future studies.

## 5. Conclusions

This study found that some anthropometric features predict soleus muscle depth, as assessed with ultrasound, in asymptomatic individuals with proper accuracy, but these features are different in men and women. Weight, height or BMI were more relevant in men, whereas calf girth and calf length were more relevant in women. Our findings could help clinicians in choosing the proper length of the needle and avoiding passing through the soleus during the application of dry needling procedures.

## Figures and Tables

**Figure 1 diagnostics-10-00284-f001:**
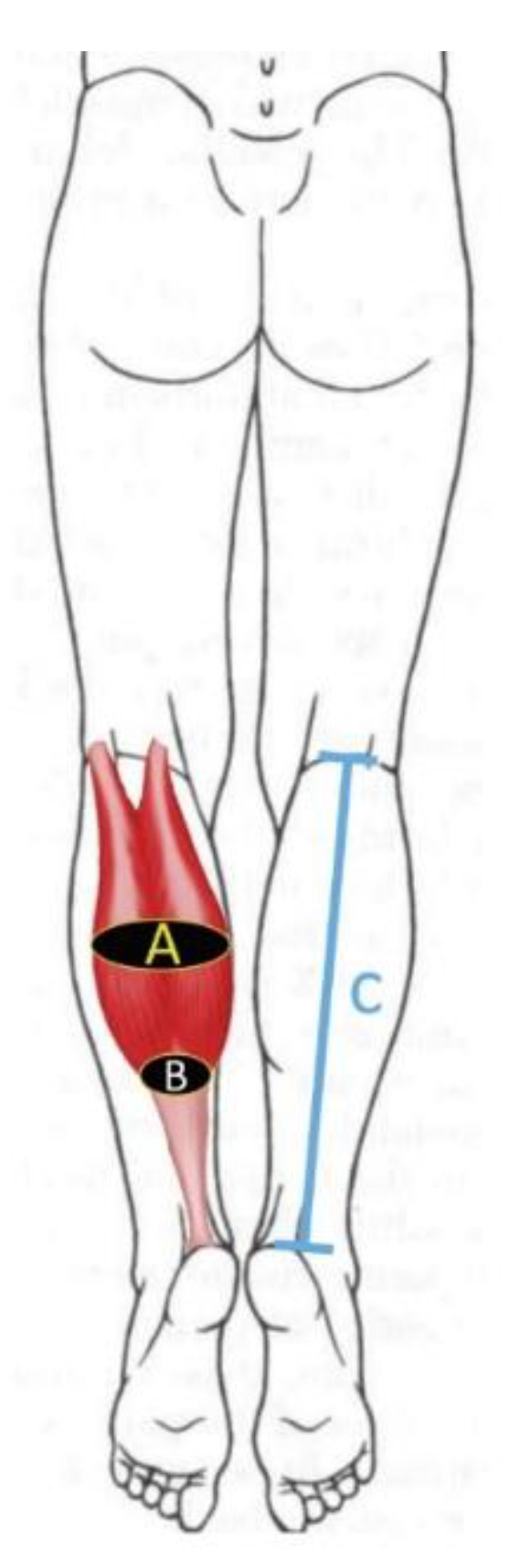
Anthropometric features assessed on the calf: (**A**) Calf girth assessed in the mid-third of the calf; (**B**) Calf girth assessed in the distal -third of the calf; (**C**) Calf length (measured from the popliteal fossa to the calcaneal tuberosity).

**Figure 2 diagnostics-10-00284-f002:**
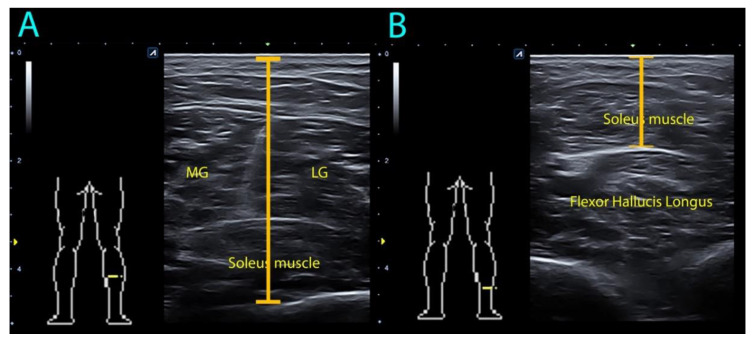
Ultrasound assessment of soleus muscle depth in the (**A**) mid-third and (**B**) distal-third of the calf. MG: medial gastrocnemius; LG: lateral gastrocnemius.

**Table 1 diagnostics-10-00284-t001:** Anthropometric features and soleus muscle depth of the total sample and by gender.

	Subjects (calves)	Age (years)	Height (m)	Weight (kg)	BMI (kg/m^2^)	Calf Length (cm)	Distal Third Calf Perimeter (cm)	Mid-Third Calf Perimeter (cm)	Distal- Third Soleus Depth (cm)	Mid-Third Soleus Depth (cm)
Sample	48	26.0 ± 7.0	1.75 ± 0.1	76.0 ± 16.0	24.0 ± 3.8	43.8 ± 2.5	25.0 ± 2.5	31.6 ± 3.1	2.2 ± 0.5	3.5 ± 0.5
**Gender**										
Male	36 (*n* = 72)	24.5 ± 5.5 *	1.80 ± 0.05 *	80.5 ± 14.0 *	24.7 ± 3.7 *	44.6 ± 2.2 *	25.0 ± 2.2	31.5 ± 2.9	2.1 ± 0.4 *	3.4 ± 0.4 *
Female	12 (*n* = 24)	30.0 ± 9.5	1.70 ± 0.05	62.0 ± 12.5	21.6 ± 3.2	41.3 ± 1.6	25.5 ± 3.2	32.0 ± 3.9	2.5 ± 0.5	4.0 ± 0.7
**Dominance**										
Dominant	24 (*n* = 48)	26.0 ±7.0	1.75 ± 0.1	76.0 ± 16.0	24.0 ± 3.8	43.6 ± 2.7	25.4 ± 2.5	32.0 ± 3.0	2.2 ± 0.5	3.6 ± 0.5
Non-dominant	24 (*n* = 48)	26.0 ±7.0	1.75 ± 0.1	76.0 ± 16.0	24.0 ± 3.8	43.9 ± 2.5	24.9 ± 2.5	31.5 ± 3.3	2.2 ± 0.5	3.5 ± 0.6

* Significant differences between men and women (*p* < 0.001).

**Table 2 diagnostics-10-00284-t002:** Pearson-Product Moment Correlation Matrix.

		1	2	3	4	5	6	7	8
Men (*n* = 72 calves)	1. Distal-third soleus depth								
	2. Mid-third soleus depth	0.597 **							
	3. Age	n.s.	n.s.						
	4. Height	0.411 **	n.s.	n.s.					
	5. Weight	0.307 **	0.272 *	n.s.	0.474 **				
	6. BMI	0.536 **	0.402 **	n.s.	n.s.	0.734 **			
	7. Calf length	−0.281 *	-0.245 *	n.s.	0.829 **	0.500 **	0.232 *		
	8. Distal-third calf girth	0.389 **	0.295 *	n.s.	n.s.	0.628 **	0.622 **	0.232 *	
	9. Mid-third calf girth	0.430 **	0.407 **	n.s.	n.s.	0.772 **	0.776 **	0.289 *	0.753 **
Women (*n* = 24 calves)	1. Distal-third soleus depth								
	2. Mid-third soleus depth	0.597 **							
	3. Age	n.s.	n.s.						
	4. Height	0.530 **	0.675 **	n.s.					
	5. Weight	0.751 **	0.861 **	n.s.	0.624 **				
	6. BMI	0.752 **	0.738 **	n.s.	0.671 **	0.773 **			
	7. Calf length	0.614 **	0.578 **	n.s.	0.723 **	0.645 **	0.777 **		
	8. Distal-third calf girth	0.719 **	0.752 **	n.s.	0.537 **	0.704 **	0.728 **	0.624 *	
	9. Mid-third calf girth	0.743 **	0.738 **	n.s.	0.493 **	0.638 **	0.790 **	0.608 **	0.778 **

n.s. non-significant; * *p* < 0.05; ** *p* < 0.01.

**Table 3 diagnostics-10-00284-t003:** Summary of the Regression Analyses to determine Predictors of Soleus Depth in Men.

	Predictor Outcome	*B*	SE B	95% CI	β	t	P
Distal-third soleus depth	Step 1						
	BMI	0.536	0.110	0.035, 0.078	0.056	2.631	0.0001
	Step 2						
	BMI	1.950	0.023	0.159, 0.251	0.205	9.931	0.0001
	Weight	1.515	0.006	0.030, 0.053	0.041	6.937	0.0001
	Step 3						
	BMI	7.165	0.118	0.520, 0.988	0.754	6.417	0.0001
	Weight	7.402	0.034	0.134, 0.271	0.203	5.890	0.0001
	Height	2.148	0.031	0.085, 0.208	0.146	4.740	0.0001
	Step 4						
	BMI	7.373	0.113	0.551, 1.001	0.776	6.889	0.0001
	Weight	7.885	0.033	0.149, 0.282	0.216	6.492	0.0001
	Height	2.285	0.030	0.096, 0.215	0.156	5.329	0.0001
	Mid-third calf girth	0.289	0.014	0.010, 0.068	0.039	2.719	0.008
Mid-third soleus depth	Step 1						
	Mid-third calf girth	0.407	0.015	0.027, 0.088	0.057	3.727	0.0001
	Step 2						
	Mid-third calf girth	0.521	0.015	0.044, 0.103	0.073	4.981	0.0001
	Calf length	−0.395	0.019	−0.108, −0.033	−0.071	−3.776	0.0001

Distal-third soleus depth: *R*^2^ adj. = 0.277 for step 1, *R*^2^ adj. = 0.568 for step 2, *R*^2^ adj. = 0.670 for step 3, *R*^2^ adj. = 0.699 for step 4; Mid-third soleus depth: *R*^2^ adj. = 0.154 for step 1, *R*^2^ adj. = 0.288 for step 2.

**Table 4 diagnostics-10-00284-t004:** Summary of the Regression Analyses to determine Predictors of Soleus Depth in Women.

	Predictor Outcome	B	SE B	95% CI	β	t	P
Distal-third soleus depth	Step 1						
	Mid-third calf girth	0.843	0.016	0.084, 0.150	0.117	7.354	0.0001
	Step 2						
	Mid-third calf girth	0.927	0.015	0.096, 0.162	0.129	8.137	0.0001
	Calf length	0.239	0.006	0.001, 0.027	0.014	2.100	0.038
	Step 3						
	Mid-third calf girth	0.969	0.005	0.044, 0.065	0.055	11.027	0.0001
	Calf length	0.459	0.006	0.013, 0.039	0.026	4.159	0.0001
	Height	0.462	0.044	0.065, 0.251	0.158	3.555	0.0002
	Step 4						
	Mid-third calf girth	0.993	0.010	0.076, 0.116	0.096	13.938	0.0001
	Calf length	0.864	0.006	0.037, 0.061	0.049	8.366	0.0001
	Height	1.365	0.064	0.333, 0.599	0.466	7.309	0.0001
	Distal-third calf girth	0.764	0.013	0.044, 0.099	0.072	5.407	0.0001
Mid-third soleus depth	Step 1						
	Mid-third calf girth	0.807	0.014	0.120, 0.195	0.158	8.620	0.0001
	Step 2						
	Mid-third calf girth	0.878	0.008	0.150, 0.210	0.180	12.833	0.0001
	Calf length	0.450	0.024	0.148, 0.249	0.198	8.272	0.0001
	Step 3						
	Mid-third calf girth	0.666	0.119	0.095, 0.144	0.119	10.261	0.0001
	Calf length	0.259	0.114	0.037, 0.192	0.114	3.081	0.0001
	Weight	0.282	0.016	0.004, 0.028	0.016	2.739	0.013

Distal-third soleus depth: *R*^2^ adj. = 0.498 for step 1, *R*^2^ adj. = 0.538 for step 2, *R*^2^ adj. = 0.632 for step 3, *R*^2^ adj. = 0.730 for step 4; Mid-third soleus depth: *R*^2^ adj. = 0.561 for step 1, *R*^2^ adj. = 0.671 for step 2, *R*^2^ adj. = 0.678 for step 3.

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
