# Peer review of "Prediction Model of Soleus Muscle Depth Based on Anthropometric Features: Potential Applications for Dry Needling"

_diagnostics, 2020, doi:10.3390/diagnostics10050284_

Round 1

Reviewer 1 Report

Thank you for the opportunity to review your paper. It is interesting, but I suggest adding more detail especially in the introduction. Please see my comments and suggestions:

Introduction

Line 43: controls ankle dorsiflexion (add:) when contracting eccentrically

Line 43-45: suggest to add this sentence somewhere in the next paragraph towards the end

Line 47: I don’t think that TrPs are one of the most common muscular impairments, please delete ‘most’ (or add a reference to support your statement)

Line 49: what kind of stimulation are you talking about?

Line 57: this reference [8] is old, is there something more recent?

Line 59: change to: which includes [the soleus muscle]

Lines 60-61: I don’t understand the connection between the two sentences that you link with ‘however’.

Line 62: please add more information on dry needling. Right now there is only wording in parentheses dedicated to the technique. This is an important detail as your study is founded on it.

Line 64: please format this reference correctly

Line 64: please change ‘is deep’ to something like ‘is located deep to the gastrocs’, or something similar

Line 65: the needle should not just reach the muscle, but also penetrate it

Line 66: please discuss in greater detail the path of the tibial nerve and how it could get injured by dry needling (maybe relate it to some dry needling methods).

Line 74-75: add ‘number of’ to “reduce {number of} adverse events”

Please add more information on the anatomy of the soleus muscle, like its thickness at different locations, where are frequent sites for TPs and therefore dry needling?

Participants

Line 101 and 102: what is the difference between ‘pain symptoms’ and ‘recurrent pain’?

Figure legend: please align with the wording you use in the text

Line 115: how did you confirm ankle dorsiflexion 0 degrees? How did you keep the ankle in that position?

Line 128: I don’t understand how you used the student’s t-test to determine gender and dominance differences

Results

I would like to see one equation that shows the model (or models). This would add clarity.

Discussion

Line 184: please delete ‘s’ on depend

Line 187: please delete ‘ing’ on avoiding

Line 190: I don’t understand ‘according to the forearm girth’; do you mean “depending’?

Line 191: please either add “the” clinician or add a “s” to ‘clinician’

Line 191-193: these sentences are not clear (what is ‘deep muscle limit’), please reword.

Line 200: delete “in” [men].

Clinical applications

If I understand your suggestions correctly, the clinician could measure the soleus’ circumference locations and the lower leg length, measure the patient’s height and weight, and calculate the BMI so he/she can determine appropriate needle length? While this is an interesting mental exercise, I just don’t think that any clinician will go through these procedures.

Author Response

We thank the reviewers for their comments, which have clarified several aspects of the manuscript. We would like to make the Editor know that we addressed all suggestions made by the reviewers. All changes are highlighted to facilitate the review process.

Thank you for the opportunity to review your paper. It is interesting, but I suggest adding more detail especially in the introduction.

Response: We would like to thank to the reviewer for the positive feedback

Please see my comments and suggestions:

Introduction

Line 43: controls ankle dorsiflexion (add:) when contracting eccentrically

Response: This sentence has been completed as follows (lines 45-47):

The soleus muscle contributes, with the gastrocnemius muscles, to ankle plantar-flexion, but also provides postural control of the ankle, particularly in weight bearing since it controls ankle dorsiflexion when acting eccentrically.

Line 43-45: suggest to add this sentence somewhere in the next paragraph towards the end

Response: We have moved this sentence to the end of the next paragraph as requested (lines 60-61).

Line 47: I don’t think that TrPs are one of the most common muscular impairments, please delete ‘most’ (or add a reference to support your statement)

Response: We have deleted the word “most” as requested.

Line 49: what kind of stimulation are you talking about?

Response: We have clarified this as follows on lines 50-52:

            A TrP is defined as a sensitive spot in a skeletal muscle taut band that is painful on stimulation (e.g., manual palpation, muscle contraction, needling), elicits referred pain, and induces motor disturbances [3].

Line 57: this reference [8] is old, is there something more recent?

Response: Reference 8 is from 2013 and it was one of the first discussing the role of TrPs in the triceps sural and ankle dorsiflexion. We have included more recent studies from the same group confirming this hypothesis, but we believe that this reference should be included since it was the first one. We believe that a reference from 2013 is relatively not “old”

Line 59: change to: which includes [the soleus muscle]

Response: Edited (lines 62-63)

Lines 60-61: I don’t understand the connection between the two sentences that you link with ‘however’.

Response: We have edited this sentence as follows (lines 64-66):

All studies reported positive results in pain and mobility outcomes with the application of manual therapy; however, it should be considered that manual access to the soleus muscle is only possible on the lateral portion due to its anatomical location.

Line 62: please add more information on dry needling. Right now there is only wording in parentheses dedicated to the technique. This is an important detail as your study is founded on it.

Response: We have included the definition of dry needling as requested (lines 67-71):

The American Physical Therapy Association (APTA) defines dry needling as “an intervention using a thin filiform needle that stimulates the skin, TrPs, muscle or connective tissue for the treatment of musculoskeletal pain disorders” [13]. This intervention consists of the insertion of a solid filiform needle into a muscle with the aim to inactivate potential TrPs [12].

Line 64: please format this reference correctly

Response: This typo has been edited, thanks

Line 64: please change ‘is deep’ to something like ‘is located deep to the gastrocs’, or something similar

Response: Edited (lines 71-72)

Line 65: the needle should not just reach the muscle, but also penetrate it

Response: We have edited the word “reach” by penetrate as requested (line 72)

Line 66: please discuss in greater detail the path of the tibial nerve and how it could get injured by dry needling (maybe relate it to some dry needling methods).

Response: We have described the path of the tibial nerve on lines 76-80 as follows:

The tibial nerve is a main branch of the sciatic nerve which arises at the apex of the popliteal fossa. When the nerve travels through the popliteal fossa, it passes deep to the tendinous arch of the soleus muscle, usually at the mid-line, to enter the posterior compartment of the calf. In the calf, the nerve runs downwards and medially to the mid-line to finally reach the posteromedial side of the ankle [14].

The topic of dry needling approaches is discussed in clinical repercussion (lines 264-271)

The results of the current study suggest that clinicians should use needles ranging from 25mm to 40mm of length for needling the soleus muscle with a posterior needle approach while keeping in mind that the muscle is deeper in women than in men. This information is clinically relevant since the tibialis posterior neurovascular package runs under the soleus muscle; therefore, using needles longer than 40mm could be risk for puncturing this muscle with a posterior approach. The current prediction model could help clinicians to identify patients for using short needles (25-30mm) based on anthropometric features. Future cadaveric studies could help to further elucidate the validity of this model in other needling approaches of the soleus muscle, e.g., medial one [12]

Line 74-75: add ‘number of’ to “reduce {number of} adverse events”

Response: Edited (lines 85-86)

Please add more information on the anatomy of the soleus muscle, like its thickness at different locations, where are frequent sites for TPs and therefore dry needling?

Response: We have expanded the anatomy of the soleus muscle on lines 41-45

“The soleus muscle originates from the soleal line and middle third of the posterior aspect of the tibia, the proximal fourth third of the posterior surface of the fibula, and the posterior aspect of the fibular head and inserts, with the gastrocnemius, into the posterior part of the calcaneus forming the Achilles tendon [1]. Chow et al reported a 2cm mean thickness of the soleus muscle, with males exhibiting greater muscle thickness in general [2].”

 Participants

Line 101 and 102: what is the difference between ‘pain symptoms’ and ‘recurrent pain’?

Response: As it is stated in the methods section (lines 109-114), subjects were included if they did not experience any symptoms of pain the previous year. However, one can understand that people could have pain symptoms more than a year ago, hence, we also excluded people with recurrent symptoms. For instance, it would be possible that a participant did not present with symptoms during the previous year, but had a history of recurrent pain some years ago. We have slightly edited the first sentence for clarification of this (lines 110-111)

To be eligible to participate, volunteers had to be between 18 and 55 years old and not report pain in the lower extremity the previous year. Exclusion criteria included prior history of recurrent pain in the lower extremity; any pharmacological treatment affecting muscle tone, e.g., muscle relaxants, analgesics; prior history of lower extremity surgery; lumbar radiculopathy or myelopathy; and any medical condition such as tumour or fracture.

Figure legend: please align with the wording you use in the text

Response: We have aligned the wording of the text to the figure legend.

Line 115: how did you confirm ankle dorsiflexion 0 degrees? How did you keep the ankle in that position?

Response: We are very sorry, since we did not include this detail in the previous version of the text. We have now clarified this on lines 127-128:

A rigid table was placed on the foot surface of participants for maintaining the ankle position

Line 128: I don’t understand how you used the student’s t-test to determine gender and dominance differences

Response: As it is stated in the statistical analysis section (line 148-149), we used student t-test to analyse gender (dominance) differences in mean values of the data. Since there are two different groups (male/female or left/right), this is the appropriate test.

Results

I would like to see one equation that shows the model (or models). This would add clarity.

Response: We are very sorry, but we do not understand what the reviewer is asking for. We have included all the data from the logistic models in the tables as  is commonly presented in scientific papers.

Discussion

Line 184: please delete ‘s’ on depend                   

Response: Edited (line 223)

Line 187: please delete ‘ing’ on avoiding             

Response: Edited (line 226)

Line 190: I don’t understand ‘according to the forearm girth’; do you mean “depending’?

Response: Edited (line 229)

Line 191: please either add “the” clinician or add a “s” to ‘clinician’

Response: Edited (line 230)

Line 191-193: these sentences are not clear (what is ‘deep muscle limit’), please reword.

Response: We have edited this sentence as requested on lines 231-233

This could result in needling approaches not considering muscle depth, such as of the soleus, which could lead to an accidental nerve puncture of deeper neurovascular structures.

Line 200: delete “in” [men].

Response: Edited (line 239)

Clinical applications: If I understand your suggestions correctly, the clinician could measure the soleus’ circumference locations and the lower leg length, measure the patient’s height and weight, and calculate the BMI so he/she can determine appropriate needle length? While this is an interesting mental exercise, I just don’t think that any clinician will go through these procedures.

Response: Since age, gender, dominance, height and weight data are usually registered on the clinical interview, we believe that clinicians would not need more than 3min to assess calf length and girth in a patient. In absence of an US equipment, these 3 minutes could assist in a more precise needling intervention. We have included a brief comment on this on line 263:

Nevertheless, calculation of all anthropometric features could be slightly time consuming.  

Reviewer 2 Report

The manuscript presents an interesting topic regarding the soleus muscle depth and development of a prediction model using anthropometric characteristics. This study can have a clinical application for dry needling procedure. However, I have some comments.

Introduction. I think that the use of term "calf" instead of leg would not lead to any confusion. This recommendation concerns the entire manuscript.

Soleus muscle controls the plantarflexion of the foot (see lines 42-43).

Methods.

Explain how was the sample size calculated.

Remove the number of recruited subjects (line 98) as this is mentioned in the Results.

How was the procedure regarding the local announcements conducted?

Conclusions. Include what anthropometric features predict the soleus muscle depth.

Author Response

We thank the reviewers for their comments, which have clarified several aspects of the manuscript. We would like to make the Editor know that we addressed all suggestions made by the reviewers. All changes are highlighted to facilitate the review process.

The manuscript presents an interesting topic regarding the soleus muscle depth and development of a prediction model using anthropometric characteristics. This study can have a clinical application for dry needling procedure.

Response: We would like to thank the reviewer for this positive feedback

However, I have some comments.

Response: We hope that we have positively answered all comments.

Introduction. I think that the use of term "calf" instead of leg would not lead to any confusion. This recommendation concerns the entire manuscript.

Response: We have edited the term leg with calf in the text as requested

Soleus muscle controls the plantarflexion of the foot (see lines 42-43).

Response: This sentence has been completed as follows (lines 45-47):

The soleus muscle contributes, with the gastrocnemius muscles, to ankle plantar-flexion, but also provides postural control of the ankle, particularly in weight bearing since it controls ankle dorsiflexion when acting eccentrically.

Methods.

Explain how was the sample size calculated.

Response: We have included the original explanation on lines 137-144 as requested

2.4. Sample size Calculation

Based on the only previous similar study [20], a sample size of at least 65 calves could be considered as appropriate. If we considered our design as a prognostic study, a range from 10 to 15 subjects per potential predictor, with no more than 5 predictor variables, is usually recommended to develop an adequate sample size for prediction models and for avoiding overestimation of the results [23]. Based on this alternative calculation a sample size of 50 calves would be required given the maximum cut-off of 5 predictors included in the final model. Since we originally include eight potential anthropometric features in the model, we calculated a sample size of at least 80 calves.

Remove the number of recruited subjects (line 98) as this is mentioned in the

Response: The first paragraph of this section has been edited as follows (lines 108-109)

Consecutive healthy volunteers recruited from local announcements located at a private clinic of Alfonso X El Sabio University between January-February 2020 were invited to participate

Results.

How was the procedure regarding the local announcements conducted?

Response: We are very sorry, but we do not understand this question from the reviewer. We put an announcement out and people interested in the study was interviewed for potential eligible criteria, as it is commonly done in many studies. As it is stated in the first line of the results (lines 166-168), from 50 volunteers who responded to the announcement, two were excluded. We have clarified that we put flyers on lines 108.

Conclusions. Include what anthropometric features predict the soleus muscle depth.

Response: We have included the following sentence on lines 282-283

Weight, height or BMI were more relevant in men, whereas calf girth and calf length were more relevant in women.

We hope that the current version of the manuscript can get a positive review and can be accepted for publication in Diagnostics

Sincerely yours,

The authors

Round 2

Reviewer 2 Report

The authors addressed all my comments.